# Guts Bacterial Communities of *Porcellio dilatatus*: Symbionts Predominance, Functional Significance and Putative Biotechnological Potential

**DOI:** 10.3390/microorganisms10112230

**Published:** 2022-11-11

**Authors:** Catarina Coelho, Igor Tiago, António Veríssimo

**Affiliations:** Centre for Functional Ecology, Department of Life Sciences, University of Coimbra, 3000-456 Coimbra, Portugal

**Keywords:** metagenome, gut bacterial communities, *Porcellio dilatatus*, LCB-degrading enzymes, biodegradation pathways, biocatalysts

## Abstract

Terrestrial isopods are effective herbivorous scavengers with an important ecological role in organic matter cycling. Their guts are considered to be a natural enrichment environment for lignocellulosic biomass (LCB)-degrading bacteria. The main goal of this work was to assess the structural diversity of *Porcellio dilatatus* gut bacterial communities using NGS technologies, and to predict their functional potential using PICRUSt2 software. Pseudomonadota, Actinomycetota, Bacillota, Cyanobacteria, Mycoplasmatota, Bacteroidota, *Candidatus* Patescibacteria and Chloroflexota were the most abundant phyla found in *P. dilatatus* gut bacterial communities. At a family level, we identified the presence of eleven common bacterial families. Functionally, the *P. dilatatus* gut bacterial communities exhibited enrichment in KEGG pathways related to the functional module of metabolism. With the predicted functional profile of *P. dilatatus* metagenomes, it was possible to envision putative symbiotic relationships between *P. dilatatus* gut bacterial communities and their hosts. It was also possible to foresee the presence of a well-adapted bacterial community responsible for nutrient uptake for the host and for maintaining host homeostasis. Genes encoding LCB-degrading enzymes were also predicted in all samples. Therefore, the *P. dilatatus* digestive tract may be considered a potential source of LCB-degrading enzymes that is not to be neglected.

## 1. Introduction

Terrestrial isopods (Crustacea: Isopoda), commonly known as woodlice, are ubiquitous in soil and constitute effective herbivorous scavengers [1]. They are considered keystone species in terrestrial ecosystems due to their crucial role in organic matter decomposition [2]. They have an important ecological role in organic matter cycling through the fragmentation of the substrate, the promotion of hindgut lignocellulosic biomass (LCB) decomposition by LCB-degrading microbial populations, and the dissemination of those microbes throughout terrestrial ecosystems [3]. Lignocellulosic biomass is the most abundant renewable biomass source on Earth, and it has been considered an important source of bioenergy or other added value products [4]. It is a heterogenous complex matrix, essentially composed of carbohydrate polymers of cellulose, hemicellulose and aromatic polymers of lignin [5]. Cellulose is a homopolymer only composed of linear chains of glucose units linked by β-(1,4)-glycosidic bonds. Its enzymatic hydrolysis specifically involves the synergetic action of three different classes of glycoside hydrolases: endoglucanases (EC 3.2.1.4), exoglucanases (or cellobiohydrolases) (EC 3.2.1.91) and β-glucosidases (EC 3.2.1.21) [6,7]. Hemicellulose is a heteropolymer composed of several basic sugars, like xylose, mannose, which is often branched with arabinose, galactose, and other acidic sugars [2,8]. Its enzymatic hydrolysis requires the synergetic action of three classes of enzymes: endo-hemicellulases, exo-hemicellulases, and accessory enzymes (or debranching enzymes) [2]. Lignin is an aromatic heteropolymer, composed of phenylpropanoid aryl-C3 units linked via a variety of ether and carbon–carbon linkages [2]. Its enzymatic degradation involves the action of two main groups of enzymes, designated by the lignin-modifying enzymes (LME) and lignin-degrading auxiliary (LDA) enzymes. The LME are enzymes directly involved in lignin degradation, like the manganese-dependent peroxidases (EC 1.11.1.13), lignin peroxidases (EC 1.11.1.14), and laccases (EC 1.10.3.2). Conversely, the LDA (or accessory) enzymes such as aryl alcohol oxidase (EC 1.1.3.7), cellobiose dehydrogenase (EC 1.1.99.18) and glucose oxidase (EC 1.1.3.4) are unable to degrade lignin on their own, but they are necessary to complete the degradation process [2,9,10,11]. Commercially, the complex composition and recalcitrant structure of LCB has difficulted its degradation and valorisation. So, for an efficient and more complete lignocellulosic biomass degradation, will be necessary the synergetic and cooperative action of different classes of enzymes. For example, hydrolytic enzymes able to metabolizable carbon sources like cellulose, hemicellulose, arabinose, starch and oxidative enzymes like peroxidases, laccases, or other auxiliary enzymes can catalyze the oxidation of the aromatic compounds [12,13,14].

*Porcellio dilatatus* is an abundant terrestrial isopod in the Atlantic Islands, Iberian Peninsula and North African countries [15] and is considered an important member of the arthropod communities inhabiting the upper layer of the soil and surface leaf litter [16]. In contrast with *Porcellio scaber* and *Armidillium vulgare*, *P. dilatatus* is a less cosmopolitan species and, consequently, less studied in research works about terrestrial isopods. Apart from ecotoxicological studies, most of the research about terrestrial isopods is related to symbiotic associations of endosymbiotic bacteria like *Wolbachia* [17,18], the bacterial pathogen *Rickettsiella* [19] and hepatopancreatic colonizing Mycoplasma-like symbiont such as *Candidatus Hepatoplasma* [20,21]. Briefly, *Wolbachia* sp. are obligate intracellular bacteria widespread in arthropods and also found in nematodes [22,23,24]. These endosymbionts can be vertically transmitted and work as reproductive parasites, manipulating the host reproduction by increasing the feminization of the isopods, being involved in parthenogenesis and male-killing events [17,23]. Occasionally, *Wolbachia* species can also be transmitted among distinct species, causing the widespread distribution of this bacteria in different invertebrate hosts [17,23,25]. *Rickettsiella* sp. are well-known intracellular pathogens found in a wide range of hosts like insects, crustaceans, and arachnids [3,26]. In 1970, Vago et al. [27] identified this pathogen in *Armadillidium vulgare* isopods, and since then *Rickettsiella* had been detected in other terrestrial isopods such as *Porcellio dilatatus* [28] and *Porcellio scaber* [29,30]. The *Candidatus Hepatoplasma* is a known bacterial symbiont specifically associated with midgut glands of terrestrial isopods [3]. Its acquisition is considered a potential evolutionary prerequisite for terrestrial colonization by isopods [3,31], since isopods harboured *Candidatus Hepatoplasma* have an enhanced ability to survive in nutritional stress conditions [32].

In recent years, the study of host–symbiont binary interactions has been complemented with studies about the impact of the microbiome on the development and nutrition of the host, following the holobiont concept [2,33,34]. So, the available information about the microbiome composition, symbiotic relationships between terrestrial isopods and their associated microbial communities [35,36], and the functional role of their microbial communities is still limited. To overcome this gap, and since the terrestrial arthropod guts were described as important, “hot-spots” for microbial-cellulolytic degrader populations [37], as well as a high bacterial diversity and several LCB-degrading enzymes of prokaryotic origin, were identified in *Armadillium vulgare* digestive tissues by metagenome and shotgun approaches [2,34,38]. We find it very believable that the terrestrial isopods, due to their know feeding habits and dietary composition, also could have a high LCB-degrading potential.

In the present work, we reported a detailed analysis of the *P. dilatatus* gut bacterial communities’ composition, which we had assessed by 16S rRNA gene amplicon sequencing, and predicted its functional profile using PICRUSt2 (Phylogenetic Investigation of Communities by Reconstruction of Unobserved States 2) software. The principal objectives of the present work were: (1) to assess the structural bacterial diversity present in several *P. dilatatus* gut samples; (2) to understand the potential functional role of gut bacterial communities to explore host–microbe interactions; and (3) to evaluate the potential presence of LCB-degrading enzymes in *P. dilatatus* gut bacterial communities.

## 2. Materials and Methods

### 2.1. Sample Collection

*Porcellio dilatatus* specimens were collected from different sites in Portugal during the Autumn of 2019: from the Mértola region (southeastern of Portugal), the Cabeço-de-Vide region (reference PdCV, located at southeast of Portugal, 39°8′1.277″ N, 7°34′44.472″ W) and the Botanical Garden of University of Coimbra (reference PdBT, located at centre of Portugal, 40°12′22.493″ N, 8°25′26.778″ W). In the Mértola region, isopods were collected from two sites—Tronco (reference PdTr, coordinates 37°40′56.8″ N, 7°30′42.2″ W) and Corte-Pinto (reference PdCP, coordinates 37°34′28.3″ N, 7°57′47.8″ W). After collection, and to preserve the gut content, isopods were stored in plastic boxes and kept on ice in coolers during transportation to a laboratory, where the animals were washed with sterile distilled water to remove many contaminating surface bacteria and kept at −70 °C until DNA extraction.

### 2.2. Gut Isolation and DNA Extraction

Three to seven isopods per sampling site (for a total of twenty-two individuals) were washed with sterile distilled water and their guts were then aseptically extracted and processed, as previously described by [39]. Heads and legs were removed with tweezers and scalpel under magnifying glass. The guts (from the esophagus to the rectum, without hepatopancreas) were separated from the rest of the digestive tract and cut longitudinally with a scalpel. Guts were placed separately in 1.5 mL tubes containing 500 µL of phosphate-buffered saline (PBS) solution pH 7.2 [40]. Then, they were vortexed, and the suspension was used to extract total DNA using Powersoil DNA isolation Kit (MO BIO Laboratories Inc., CA, USA) following the manufacturer′s instructions. DNA concentration and quality were determined using Nanodrop (Thermo Scientific, Waltham, MA, USA) and 1% agarose gel electrophoresis, respectively.

### 2.3. rRNA Gene Sequencing and Data Analysis

Twenty-two samples of gut genomic DNA were used to conduct the bacterial 16S rRNA gene amplicon sequencing in an Illumina MiSeq V2 platform in accordance with the sequencing facility provider and manufacturer protocols (Marine Biological Laboratory at Woods Hole, USA). The structural diversity of the communities was determined through the amplification of hypervariable regions V1-V3 of the bacterial 16S rRNA gene using the primers 28F-GAGTTTGATCNTGGCTCAG-, 519R–GTNTTACNGCGGCKGCTG-. Raw data were analyzed using mothur v.1.47 software package (http://www.mothur.org (accessed on 20 May 2020); [41]). Briefly, sequences were subjected to conservative quality control measures at initial quality trimming and assembly of contig reads sequences, before classification, using the command: screen.seqs(fasta = current, count = current, maxambig = 0, maxlength = 550, minlength = 300, processors = 7, maxhomop = 9). This removed all sequence reads with low quality and ambiguous bases from the data sets. Sequences representing chimeras were removed with the command chimera.vsearch (fasta = current, count = current, dereplicate = t). The resultant sequences were aligned and clustered into operational taxonomic units (OTUs) with a cut-off of 0.03 (97% similarity). All high-quality sequences were phylogenetically identified through ARB-Silva taxonomic database version 138 [42,43]. All OTUs taxonomically classified as chloroplast, archaea, eukaryota and mitochondria were removed from the dataset. Inverse Simpson, evenness, and Shannon evenness diversity indexes, the Chao richness values, coverage and rarefaction curves were generated using mothur v.1.47 software. The bacterial 16S rRNA gene Illumina sequencing data were deposited in the NCBI BioProject library (BioProject ID PRJNA885808).

### 2.4. Functional Analysis

The metabolic profiles of *P. dilatatus* gut bacterial communities were predicted using the PICRUSt2 software employing the Kyoto Encyclopedia of Genes and Genomes (KEGG) database [44]. Predicted functions were assigned to KEGG pathways up to the hierarchical fourth level. “Metabolism” was the functional module on which we focused our analyses. Furthermore, we used PICRUSt2 results at the KEGG-fourth level to predict the presence of genes encoding LCB-degrading enzymes, selected by their recognized activity. The selection of genes encoding LCB-degrading enzymes was based on CAZy database (http://www.cazy.org/ (accessed on 15 February 2021)) and previous scientific works about LCB-degrading enzymes [4,9,14]. The accuracy of the predicted metabolic profiles for each sample was assessed using the weighted nearest sequenced taxon index (NSTI) provided by PICRUSt2 software. This meant that only OTUs with an NSTI value below 2 were used to determine the metabolic profiles, while OTUs above NSTI 2 were discarded from the analyses since they had no close relative with genomic information available. This ensured the accuracy of the predictions obtained.

### 2.5. Statistical Analysis

The comparison between the structural bacterial diversity present in the different *P. dilatatus* gut samples was performed using principal coordinate analysis (PCA). For visual assessment of the functional diversity distribution among the different samples, a heatmap was generated. The previous graphs were constructed and analyzed by ANOVA and Tukey–Kramer post hoc tests using the Statistical Analysis of Metagenomic Profiles (STAMP) software [45]. The alpha-diversity analysis and core microbiome determination of the 16S rRNA gene abundance data was performed on MicrobiomeAnalyst (v.4.1.3) (https://www.microbiomeanalyst.ca/ (accessed on 30 October 2022), [46]), an interactive web-based platform for microbiome analysis that runs on R script and provides well-established tools for microbiome data processing, statistical analysis, functional profiling and even comparison with public datasets [46,47]. A low count and variance filter was applied to the data to remove low-quality and/or uninformative features that could be associated with sequencing errors or low-level contamination. Additionally, samples with small library sizes were excluded from the analysis and a Shapiro–Wilk normality test was performed on R (v.4.2.0) for all remaining samples with the package “dplyr”. For data visualization, the comparison between the alpha diversity index obtained for all samples was accessed with the observed OTUs values and a statistical method with a non-parametric Mann–Whitney/Kruskal–Wallis test. The core microbiome was determined at a family level by considering the set of taxa detected at a minimum sample prevalence of 50% and with a relative abundance of the populations above 0.01%.

## 3. Results

### 3.1. Sequencing Data Analysis

The *Porcellio dilatatus* gut genomic DNA was assessed by 16S rRNA gene amplicon sequencing, and a total of 1,002,009 sequences were produced. After quality control, 834,714 sequences were selected for further analysis. The number of high-quality sequences obtained for each sample is shown in Appendix A. All high-quality sequences were phylogenetically identified, from phylum to genus. From a total of 9072 OTUs obtained, only 3293 OTUs were represented by ≥10 sequences in at least one sample, and their 794,279 sequences were assigned to 17 different bacterial phyla, 45 classes, and 225 families, and only 0.8% of sequences (55 OTUs) could not be assigned to any known phyla. Despite differences in the number of sequences obtained from each sample, a tendency to saturation was observed in the rarefaction curves (Appendix A), demonstrating that our analysis accurately described the bacterial diversity present in *P. dilatatus* gut samples. Moreover, the calculated coverage values reinforced this observation (Appendix A). The bacterial diversity present in each *P. dilatatus* gut sample was assessed by alpha-diversity indexes (inverse Simpson, Shannon, Shannon evenness indexes) and richness index (Chao) shown in Appendix A. Briefly, the lowest inverse Simpson values were obtained for samples PdTr3 (1.11) and PdTr6 (2.00) representing those with lower diversity, while the highest value was obtained for samples PdBT5 (212.47) and PdBT3 (76.90) those with higher diversity. The Shannon evenness value showed that the distribution of the sequences for the different OTUs was independent from the sampling site. Most of the samples had values ranging from 0.43 and 0.77, as the lowest value was obtained for sample PdTr3 with 0.07, while the highest value was obtained for sample PdBT5 with 0.83. For the Chao index, the lowest richness value was observed for sample PdCV2 (88) and the highest richness value was observed for sample PdCP5 (2952). In addition, we could also observe that apparently the sampling sites do not influence the richness of the samples, since in all sites samples were obtained with high and low values (Appendix A). Thus, these alpha-diversity results obtained tended to show that each *P. dilatatus* gut sample shows an intra-species diversity independent of the sampling site.

### 3.2. Bacterial Communities Associated to Porcellio dilatatus Guts

From seventeen distinct phyla identified in the *Porcellio dilatatus* guts sampled, sequences classified at the phyla level as belonging to Actinomycetota, Pseudomonadota and Bacillota were present in all guts analyzed, comprising 87.4% of the sequences (2683 OTUs). Representatives of phyla Cyanobacteria (48 OTUs), Mycoplasmatota (20 OTUs), Bacteroidota (138 OTUs), *Candidatus* Patescibacteria (67 OTUs), and Chloroflexota (108 OTUs), comprising 11.3% of sequences, were identified in all sampling sites in at least one isopod gut sample. The sequences assigned to these eight phyla comprised almost 99% of the OTUs identified in *P. dilatatus* bacterial communities (Figure 1).

Despite some differences in the bacterial diversity of the isopod gut samples, 95.8% of the sequences were assigned to classes Actinobacteria, Gammaproteobacteria, Alphaproteobacteria, Oxyphotobacteria, Mollicutes, Bacteroidia, Thermoleophilia, Bacilli, and Acidimicrobiia (Appendix A). At a lower level, the most abundant families identified encompassed representatives of Diplorickettsiaceae (17 OTUs, 10.8% of total sequences), Enterobacteriaceae (69 OTUs, 8.1%), Anaplasmataceae (4 OTUs, 8.0%), Micromonosporaceae (180 OTUs, 7.6%), Microbacteriaceae (132 OTUs, 5.2%), Streptomycetaceae (51 OTUs, 4.5%), Entomoplasmatales *incertae sedis* (12 OTUs, 4.0%), Flavobacteriaceae (81 OTUs, 3.4%), Nocardiodaceae (192 OTUs, 3.2%), Pseudonocardiaceae (170 OTUs, 3.2%), Geodermatophilaceae (76 OTUs, 2.7%), Beijerinckiaceae (73 OTUs, 2.6%), Burkholderiaceae (65 OTUs, 2.2%), Rhizobiaceae (59 OTUs, 2.2%), Xanthobacteraceae (47 OTUs, 1.8%), Micrococcaceae (17 OTUs, 1.6%) and Rhodobacteraceae (131 OTUs, 1.5%) (Figure 2). Overall, these seventeen families comprised 72.7% of total sequences, while the other remaining representatives were distributed in small numbers by multiple taxa.

Among these seventeen most abundant families, a high relative abundance of sequences associated with Anaplasmataceae, Diplorickettsiaceae, and Entomoplasmatales *incertae sedis* was observed. The high predominance of these three families was due to the presence of *Wolbachia*, *Rickettsiella*, and *Candidatus Hepatoplasma* genera, respectively. Although a high relative abundance of sequences had been associated with these three genera, *Wolbachia* was represented by 4 OTUs, *Rickettsiella* was represented by 17 OTUs, and *Candidatus Hepatoplasma* by 12 OTUs, but for all only 1 OTU encompassed the majority of the classified sequences. Apart from these three families, the remaining most abundant fourteen families were identified in all sampling sites in at least one isopod, and they encompassed 61.3% of the abundance of sequences present in *P. dilatatus* guts.

As shown in Figure 3, it was possible to identify eleven common bacterial families in *Porcellio dilatatus* gut samples, namely families Anaplasmataceae, Microbacteriaceae, Enterobacteriaceae, Micromonosporaceae, Nocardioidaceae, Beijerinckiaceae, Geodermatophilaceae, Burkholderiaceae, Streptomycetaceae, Xanthobacteraceae and Pseudonocardiaceae.

To compare the bacterial population composition of the different samples, a principal coordinate analysis (PCA) using relative abundance of sequences identified at the family level was performed. Of the twenty-two gut samples analyzed, nineteen samples showed a high relatedness in the abundance and composition of their host-associated bacterial populations (Figure 4), while the remaining three samples showed high divergences due to discrepancies in the relative abundance of the bacterial populations. Indeed, in each of these samples, a clear dominance of a given population (representatives of Diplorickettsiaceae for PdTr3 and PdTr6 and representatives of Enterobacteriaceae for PdBT2) was verified (Figure 2).

### 3.3. Presence of Endosymbionts and Parasites in Porcellio dilatatus Bacterial Communities

The presence of *Wolbachia* sp., *Rickettsiella* spp. and *Candidatus Hepatoplasma* was observed in the vast majority of the *P. dilatatus* gut samples studied. Of the 22 isopods, only three collected at Cabeço-de-Vide were not infected by *Wolbachia* sp., ten were infected by *Rickettsiella* spp., and twelve were infected by *Candidatus Hepatoplasma*. Curiously, only two isopods (PdCV1 and PdCV3 collected at Cabeço-de-Vide) were not infected by *Wolbachia* sp., *Rickettsiella* spp. and/or *Candidatus Hepatoplasma*, while the other isopods collected at the same sampling site was only infected with *Rickettsiella* spp. Concerning the isopods infected by *Wolbachia*, only two were exclusively infected by *Wolbachia* (PdCP2 and PdBT1), while the remaining seventeen isopods were also infected by *Candidatus Hepatoplasma* and/or *Rickettsiella* spp. (Figure 5).

### 3.4. Prediction of Functional Diversity of the Gut Bacterial Communities

To explore the putative metabolic functions of *Porcellio dilatatus* bacterial communities, the functional gene contents of the bacterial communities present in 22 gut samples were predicted using PICRUSt2 software based on their 16S rRNA gene dataset. The relative abundance of the functional genes predicted in the different *P. dilatatus* gut samples are presented in Figure 6, while the analysis between sampling sites of predicted genes at level-3 KEGG is presented in Figure 7. The average of NSTI values were 0.18 ± 0.13 for Tronco samples; 0.18 ± 0.08 for PdCP samples; 0.16 ± 0.04 for PdCV samples and 0.15 ± 0.04 for PdBT samples. Despite the relatively high average NSTI values, it was possible to depict the major putative functions in the gut isopod bacterial communities. These NSTI values were not completely unexpected, since the most annotated genomes are related to humans, and the availability of reference genomes from these habitats/environments is limited [48]. At the same time, these NSTI values may also mean that environmental samples have a high degree of unexplored genomes [49].

The overall functional profiling (level-1 KEGG analysis) of the *P. dilatatus* gut bacterial communities showed the predicted presence of 5695 functional KEGG orthologues (KO), distributed by six functional modules. “Metabolism” was the most represented functional module, accounting for more than 50% of the entire set of all samples. Within “metabolism” (Figure 6), the majority of predicted level-2 KEGG pathways were related to “carbohydrate metabolism” (24.6–25.5%), “amino acid metabolism” (16.5–18.2%), “energy metabolism” (10.9–11.3%) and “metabolism of cofactors and vitamins” (9.3–10.5%). Moreover, two other relevant level-2 KEGG functional modules related to “metabolism” were predicted in *P. dilatatus* gut bacterial communities: “metabolism and biodegradation of xenobiotic” (5.5–6.1%) and “biosynthesis of secondary metabolites” (3.1–4.4%).

On the other hand, a considerable number of KO related to other two level-1 KEGG functional modules were predicted in *P. dilatatus* bacterial communities. Within the “environmental processing information” (9.7–11.6%), most of the predicted KO were related to “membrane transport” and “signal transduction” pathways. Within the “cellular processes” (7.9–8.5%), the majority of KO were related to “biofilm formation” and “quorum sensing” pathways.

In a more detailed analysis (level-3 KEGG analysis), it was possible to foresee the abundance and importance of predicted genes related to “metabolism” functional modules. In “carbohydrate metabolism”, a high number of predicted KO were related to “starch and sucrose metabolism”; “amino sugar and nucleotide sugar metabolism”; “glycolysis/gluconeogenesis”; “pyruvate metabolism”; “glyoxylate and dicarboxylate metabolism”; “propanoate metabolism” and “butanoate metabolism” (Figure 7). In “amino acid metabolism”, a high abundance of KO related to the metabolism of a large variety of amino acids were also predicted (Figure 7), such as “glycine, serine and threonine metabolism”; “alanine, aspartate and glutamate metabolism”; “cysteine and methionine metabolism” and “valine, leucine and isoleucine degradation”. In terms of “energy metabolism”, most representatives of KO were related to “oxidative phosphorylation”; “carbon fixation pathways in prokaryotes”; “methane metabolism”; “nitrogen metabolism” and “sulfur metabolism” (Figure 7). In “metabolism of cofactors and vitamins” the most represented KO was related to “porphyrins and chlorophyll metabolism”; “folate biosynthesis”; “pantothenate and CoA biosynthesis”; “biotin metabolism” and “nicotinate and nicotinamide metabolism” (Figure 7).

### 3.5. Prediction of LCB-Degrading Enzymes in the Bacterial Communities

At level-4 KEGG analysis, a high abundance of KO related to genes encoding enzymes involved in the degradation of cellulose, hemicellulose, lignin, and other cello-oligosaccharides (or in general LCB-degrading enzymes) were predicted in *P. dilatatus* gut bacterial communities. As showed in Appendix A, most of the predicted KO belonged to “starch and sucrose metabolism” (Appendix A) and to “amino sugar and nucleotide sugar metabolism” (Appendix A). From all predicted LCB-degrading enzymes, we highlighted the presence of KO identified as BRITE hierarchy belonging to β-glucosidase (EC 3.2.1.21), α-amylase (EC 3.2.1.1), β-amylase (EC 3.2.1.2), cellulose-1,4-β-cellobiosidase (EC 3.2.1.91), endoglucanase (EC 3.2.1.4), xylan 1,4-beta-xylosidase (EC 3.2.1.37) and chitinase (EC 3.2.1.14). As well as some lignin modifying enzymes like peroxidase (EC 1.11.1.7), catalase-peroxidase (EC 1.11.1.21), catalase (EC 1.11.1.16).

## 4. Discussion

The main goal of this work was to assess the structural diversity of *Porcellio dilatatus* gut bacterial communities, to predict their functional potential by bioinformatic tools and to foresee their potential as a source for LCB-degrading enzymes with biotechnological applicability. Although most of the scientific studies concerning isopod gut microbial diversity were performed under artificially controlled conditions [50], the present study analyzed the bacterial diversity present in the gut of *P. dilatatus* obtained directly from the field of various sites in Portugal. The majority of the *P. dilatatus* gut samples analyzed showed high bacterial α-diversity values. Pseudomonadota, Actinomycetota and Bacillota were present in all samples and accounted for the majority of OTUs, but representatives of five other bacterial phyla (Bacteroidota, *Candidatus* Patescibacteria, Chloroflexota, Mycoplasmatota, and Cyanobacteria) were identified in the gut of at least one specimen of each sampling site. Additionally, members of phyla Pseudomonadota, Actinomycetota, Bacillota, Bacteroidota, Mycoplasmatota and Cyanobacteria identified in the present work were also found to be associated with other soil-feeding invertebrates such as termites [51], earthworms [52], and cockroaches [53] and to other terrestrial isopod species like *Armidillium vulgare* [37], *Cubaris murina* [54] and *Porcellio scaber* [50]. Furthermore, at a lower taxonomic level (family level), a common bacterial set of eleven families was identified in our samples, namely Anaplasmataceae, Microbacteriaceae, Enterobacteriaceae, Micromonosporaceae, Nocardioidaceae, Beijerinckiaceae, Geodermatophilaceae, Burkholderiaceae, Streptomycetaceae, Xanthobacteraceae and Pseudonocardiaceae. Considering the nature of their dietis comprised of materials such as dead leaves and plants debris, we can speculate about the role of these bacterial populations as LCB-degrading bacteria and their symbiotic relation with the hosts. The results of this work showed the presence of a common set of bacterial populations in the gut of the *P. dilatatus.* As observed in the principal component analysis, the bacterial communities associated with *P. dilatatus* guts were very similar for almost all samples. These results reinforced the possible presence of autochthonous bacterial communities in the isopod gut, which likely lives in a mutualistic and commensalism relationship with the host, as already reported by Kostanjšek and colleagues [35]. As well as in agreement with the hypothesis of the presence of a species-specific pattern between host species and microbial communities associated, as observed in other invertebrate species like scorpions [55], termites [51,56], supralittoral talitrid amphipods [57] and beetles [58].

With regard to the presence of genera *Wolbachia, Rickettsiella and Candidatus Hepatoplasma,* as mentioned previously these are known to be widely spread in the arthropod microbial communities [3,22,23,24,26]. In our study, all isopods collected from Tronco, Corte-Pinto and the Botanical Garden of University of Coimbra were infected by *Wolbachia*, while the isopods from Cabeço-de-Vide, only two were not infected by *Wolbachia*. This result reflected the wide distribution of *Wolbachia* sp. in *P. dilatatus* populations in Portugal. Moreover, the results achieved in the present work showed that the presence of *Wolbachia* did not represent a reduction in the abundance of many bacterial taxa, as previously reported by Dittmer and Bouchon [33]. In contrast, our observations showed a lower bacterial diversity in Cabeço-de-Vide isopods without *Wolbachia* infection, leading us to speculate that endosymbiotic bacteria abundance may not be directly associated with a decrease in bacterial diversity as stated before [58].

PICRUSt2 software was used to predict the functional profile of *P. dilatatus* metagenome and to understand the diversity, dynamics, and evolution of *P. dilatatus* bacterial communities and potential host–microbial communities’ interactions. Interestingly, the functional diversity results showed the presence of a similar pattern across all samples, suggesting that KO, identified in *P. dilatatus* gut bacterial communities, can corresponding to functions described as crucial for the host. Of six functional modules predicted in *P. dilatatus* gut bacterial communities, the most represented functional modules were “carbohydrate metabolism”; “energy metabolism”; “amino acid metabolism” and “metabolism of cofactors and vitamins”. The prominent presence of genes involved “carbohydrate metabolism” (mainly in “starch and sucrose metabolism”; “amino sugar and nucleotide sugar metabolism”; “glycolysis/gluconeogenesis”; “pyruvate metabolism”; “glyoxylate and dicarboxylate”; “propanoate metabolism” and “butanoate metabolism”) emphasized the potential role of microorganisms in providing essential nutrients to the host. On the other hand, the putative presence of genes associated with the “butanoate metabolism” and “propanoate metabolism” was also important since their primary end-products are short-chain fatty acids (like acetate, butanoate or butyrate, propionate), which are assimilated by epithelial cells and may be used as an energy source by the host [59,60]. The presence of a considerable number of KO related to “nitrogen metabolism” (namely “dissimilatory nitrate reduction”; “assimilatory nitrate reduction”; “denitrification”; “nitrogen fixation” and “nitrification” pathways) showed the potential activity of symbiotic bacteria in the gut bacterial communities to provide usable forms of nitrogen to the host, as already described by Bahrndorff and colleagues [61]. This process is crucial for the host, since the nitrogen content is essential in the dynamics of a saprophagous population [62], and the capacity to fix nitrogen is absent in eukaryotes but is widely available among bacteria [10,63]. Various KO identified were associated with the level-2 KEGG “cellular community prokaryotes”, which were especially strongly distributed by the “quorum sensing” and “biofilm formation” pathways.

Quorum sensing has been described as the mechanism used by bacteria to coordinate gene expression and cooperate with one another [64], and its pathways are involved in the regulation of bacterial cooperative activities and in diverse physiological functions, like bacterial communication between cells and the biofilm formation [65,66]. The phenotypes of “quorum sensing” in bacteria include the production of exopolysaccharides required for adhesion and biofilm formation, the production of extracellular hydrolytic enzymes, and the production of antibiotic compounds which will aid in colonization, nutrient acquisition, and defense of the host [64]. The presence of genes related to “biofilm formation” pathways may provide many advantages to bacteria, such as the improvement of its capacity to resist predation, the antibiotic effects, and the improvement of the tolerance to a variety of environmental stresses [64]. Thus, we can speculate that a host (isopod), which has a microbial communities with great potential for “biofilm formation”, will have an advantage since its microbial communities will contribute positively to its development since the microbial metabolic activities would be better protected from external stresses. Horváthová and colleagues [67] reported that the presence of biofilm in food would enhance isopod growth.

We can speculate that isopods tend to feed on woody tissues have higher biofilm content [50,68], which could lead to the colonization of their hindguts with biofilm-promoting microbial populations. In our study, Bacteroidota and Bacillota phyla were common in the isopod microbial communities and are known to be predisposed to biofilm formation [51,69] and to produce enzymes with cellulolytic activity [34,69].

Most of the PICRUSt2-predicted genes ascertained encoded enzymes in the *P. dilatatus* bacterial communities described as LCB-degrading enzymes [4,11]. Among these, we highlight the presence of gene-encoding cellulases, xylanases, α-amylases, β-glucosidases, and lignin-modifying enzymes (LME), namely laccases, peroxidases and manganese peroxidases. These results allowed us to reinforce our impression about the potential role of the bacterial communities of the digestive tract of *P. dilatatus* for efficient enzymatic digestion of the lignocellulosic biomass wastes that constitute host feeding, and we can speculate about their huge potential in industrial applications. So, considering these observations, future works could encompass the determination of the enzymatic activities of the gut bacterial populations of the isopods. For instance, this could be determined by growing bacterial consortia or isolates from isopod gut samples, in defined media bearing specific substrates such as cellulose, xylan, guaiacol and 2,6-Dimethoxyphenol for enzymatic activity screening. Another possible approach would be the sequencing of the shotgun-metagenome of the total DNA of isopods guts samples; this could then be screened for specific genes encoding potential LCB-degrading enzymes, which could be cloned and tested for their enzymatic activities.

In conclusion, this work gave us more and detailed information about the structural diversity of *P.dilatatus* gut bacterial communities, as well as about its functional potential. The taxonomic results showed the presence of the eleven common bacterial families shared by the isopod gut samples. Moreover, the functional results achieved allowed us to predict the functional profile of *P. dilatatus* bacterial communities and to foresee a putative symbiotic relationship between the isopod and its gut bacterial communities. The high abundance of predicted genes related to carbohydrate, energy, amino acid, vitamin and cofactors metabolisms present in all samples allowed us to foresee the crucial role of the bacterial communities present in the digestive tract to maintain homeostasis and nutrient uptake for the host.

In our opinion, this work will contribute to increasing our understanding of the host–microbe interactions in isopods and provides an interesting model to study the synergetic action of several enzymes to achieve complete degradation of the lignocellulosic biomass within the isopod′s digestive tract. Furthermore, it reinforces the importance of a deeper metagenomic analysis of *Porcellio dilatatus* bacterial communities to identify genes encoding novel and/or more efficient LCB-degrading enzymes relevant for industrial uses.

## Figures and Tables

**Figure 1 microorganisms-10-02230-f001:**
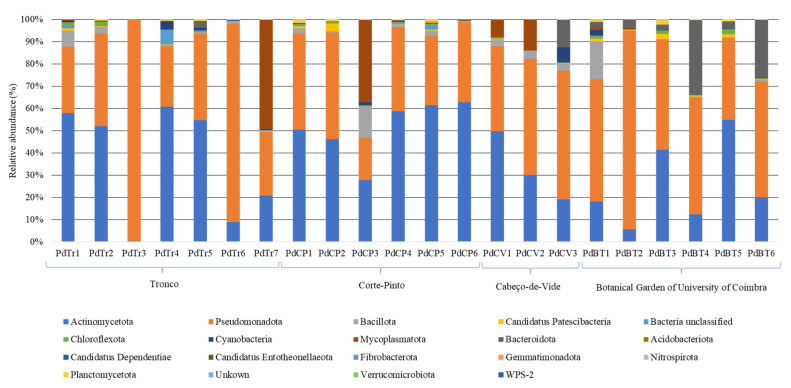
Composition of *Porcellio dilatatus* gut bacterial communities, at the phylum level. Relative abundance of sequences present in the *P. dilatatus* guts sampled from Tronco (PdTr); Corte-Pinto (PdCP); Cabeço-de-Vide (PdCV) and Botanical Garden of University of Coimbra (PdBT). Sequences that could not be classified in any known phyla were assigned as bacteria unclassified and unkown.

**Figure 2 microorganisms-10-02230-f002:**
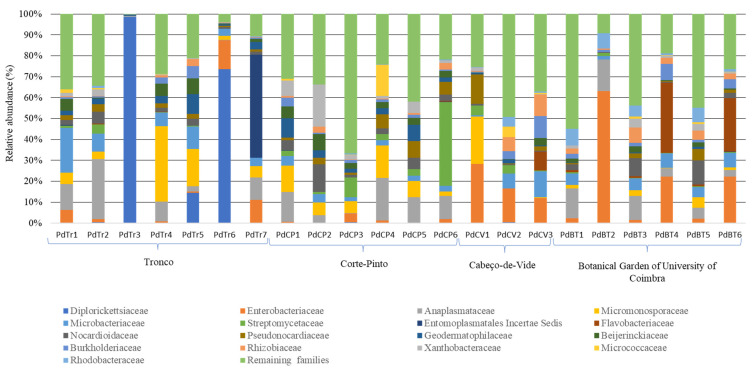
Composition of *Porcellio dilatatus* gut bacterial communities, at the family level. Isopods were collected from Tronco (PdTr); Corte-Pinto (PdCP); Cabeço-de-Vide (PdCV) and Botanical Garden of University of Coimbra (PdBT) in this study. Only the seventeen most abundant families were represented. The other families were assigned as the remaining families.

**Figure 3 microorganisms-10-02230-f003:**
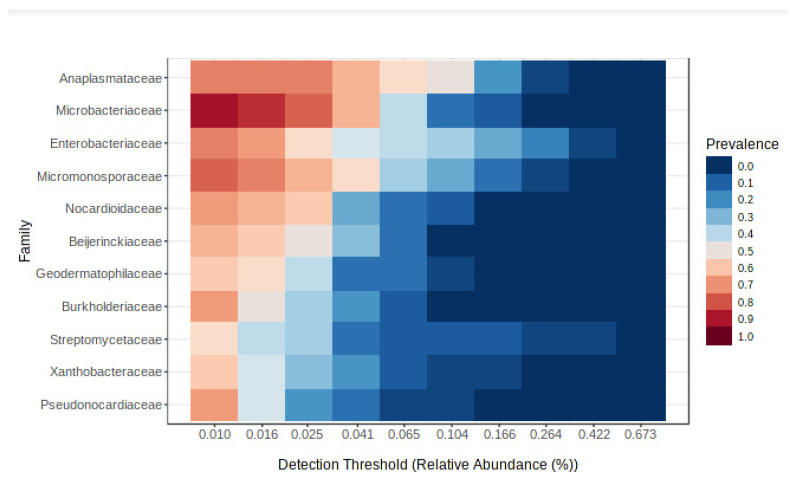
Heatmap showing the eleven common bacterial families present in *Porcellio dilatatus* gut samples. Color shading indicates the prevalence of each bacterial family among samples for each abundance threshold.

**Figure 4 microorganisms-10-02230-f004:**
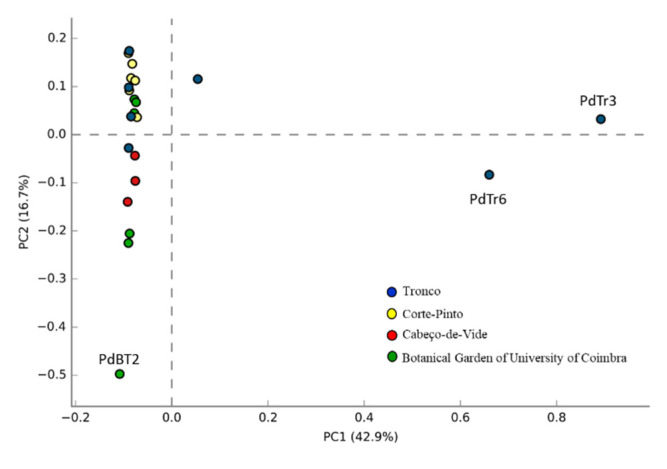
Principal component analysis of the bacteria present in *Porcellio dilatatus* gut samples, determined at the family level. *P. dilatatus* gut sampled from Tronco (PdTr) (blue); Corte-Pinto (PdCP) (yellow); Cabeço-de-Vide (PdCV) (red), and Botanical Garden of University of Coimbra (PdBT) (green). 59.6% of the variation was explained by the two principal components PC1 and PC2.

**Figure 5 microorganisms-10-02230-f005:**
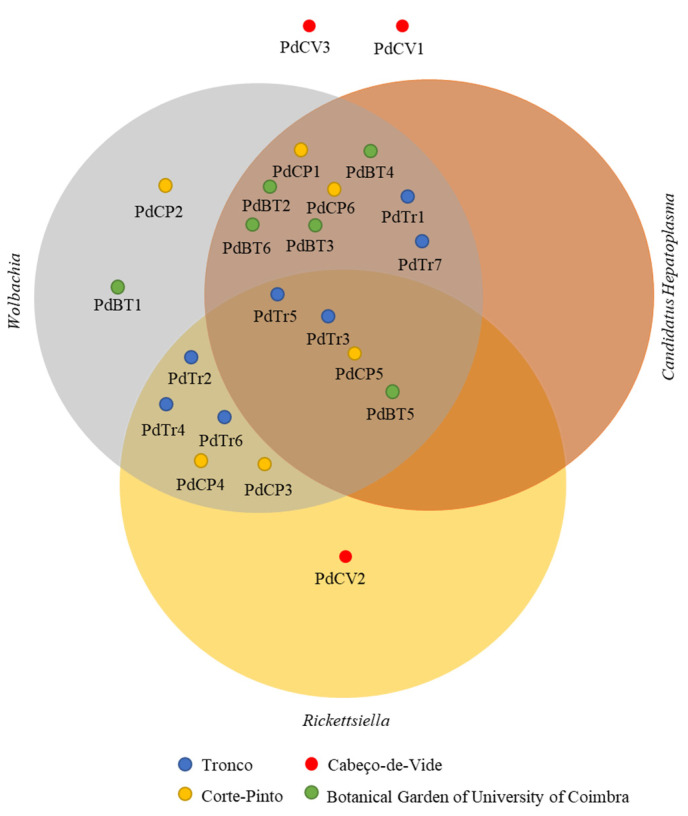
Three-way Venn diagram showing *Porcellio dilatatus* guts infected by *Wolbachia* (grey), *Rickettsiella* (yellow) and *Candidatus Hepatoplasma* (orange). *P. dilatatus* guts were sampled from Tronco (PdTr); Corte Pinto (PdCP); Cabeço-de-Vide (PdCV) and Botanical Garden of University of Coimbra (PdBT).

**Figure 6 microorganisms-10-02230-f006:**
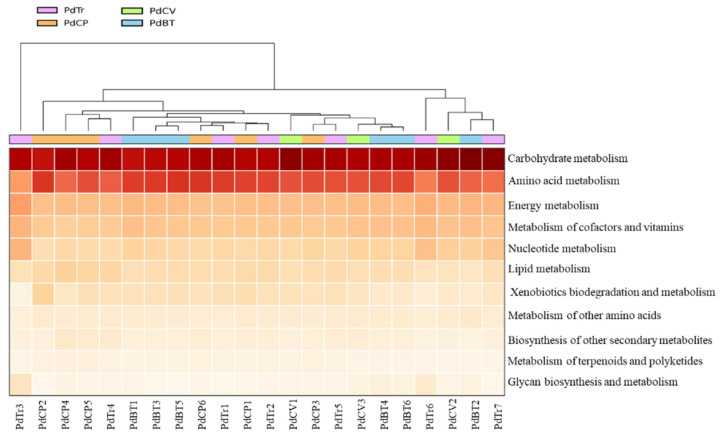
Heatmap of KEGG pathways relative abundance, per sample, associated with “metabolism” functional module predicted in *Porcellio dilatatus* bacterial communities. *P. dilatatus* guts were sampled from Tronco (PdTr) (purple); Corte-Pinto (PdCP) (orange); Cabeço-de-Vide (PdCV) (green), and Botanical Garden of University of Coimbra (PdBT) (blue).

**Figure 7 microorganisms-10-02230-f007:**
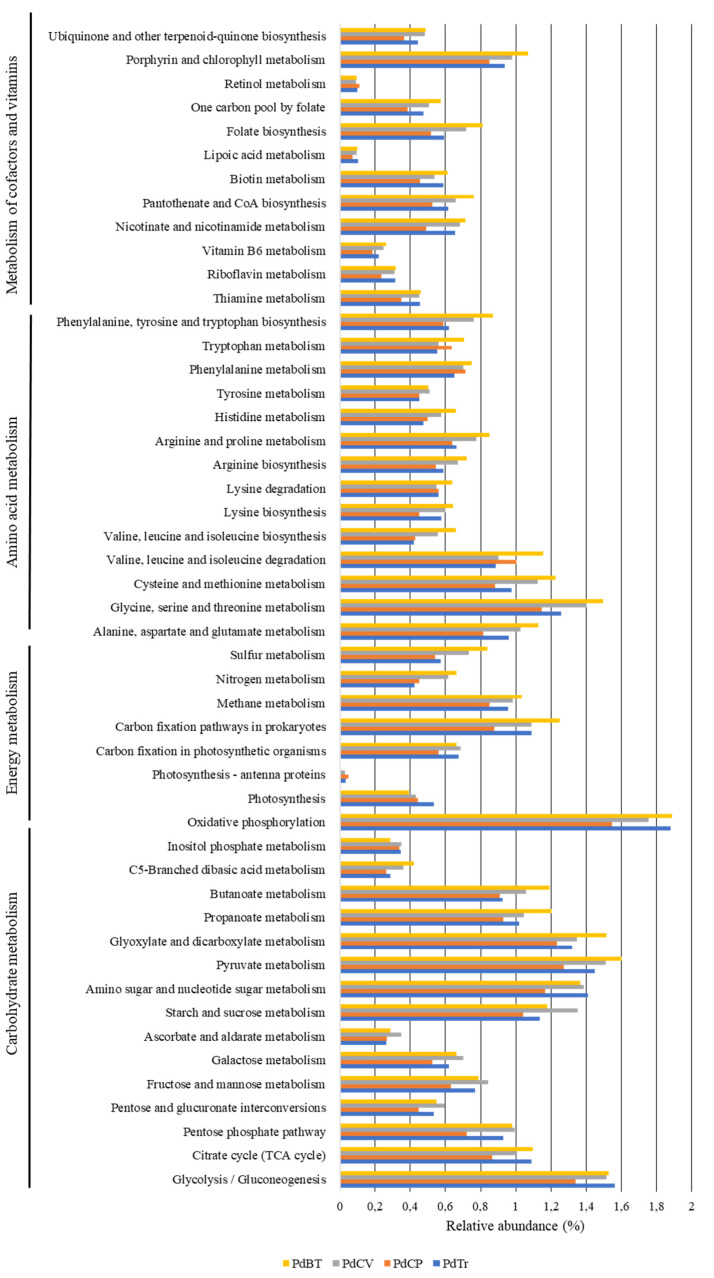
Relative abundance of KEGG pathways associated with carbohydrate metabolism, energy metabolism, amino acid metabolism, and metabolism of cofactors and vitamins in *Porcellio dilatatus* bacterial communities. *Porcellio dilatatus* gut samples from Tronco (PdTr) (blue); Corte-Pinto (PdCP) (orange); Cabeço-de-Vide (PdCV) (grey) and Botanical Garden of University of Coimbra (PdBT) (yellow).

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
