# Peer review of "Guts Bacterial Communities of Porcellio dilatatus: Symbionts Predominance, Functional Significance and Putative Biotechnological Potential"

_microorganisms, 2022, doi:10.3390/microorganisms10112230_

Round 1
Reviewer 1 Report
Please see attached doc.

Author Response
We appreciate all the comments and took them in consideration while reviewing the manuscript. Please find our replies in the attached pdf file.

Reviewer 2 Report
Summary
This manuscript provides bacterial community profiles from the gut of a woodlouse host species that has been less explored in comparison to other relatives. The data is obtained through amplicon sequencing targeting the bacterial 16S rRNA in samples collected in four different locations in Portugal. Based on this data, potential functional roles of the community members are predicted using PICRUSt2, highlighting the putative presence of digestive enzymes that are ecologically relevant for woodlice. Also, the manuscript reports on the presence and co-infection of three previously described bacteria (Wolbachia, Ricketsiella and Hepatoplasma) in the majority of the samples isopods. The data is useful as a first exploration of the microbiota associated to this species and its potential contributions to the host. I believe that some interpretations must be revised and/or clarified, and special consideration must be given to the predictive nature of the approach used for the functional aspect. Please find a few general points detailed below, as well as more specific comments noted directly on the pdf file, since no line numbering was provided.
General comments:
1. As noted in several specific comments, I suggest the authors to tone down their conclusions on function and biotechnological potential. These are predictions, meaning that there is no direct evidence of the presence of the corresponding genes, their expression or their activity in the host. Also, this approach is based on taxonomic similarity, which in many cases does not correspond to functional or ecological similarity. While the data is useful, it is important that the wording reflects the limitations of this approach.
2. Please clarify what is meant by “coverage” in Supplementary Table 1 and, in connection to this, make sure that the assessment of full representation of the diversity in the samples is fair. The manuscript states that the rarefaction curves show saturation, but this is not the case (see specific comments on pdf).
3. Including negative controls would have been an important addition to this dataset. As now generally recognized in microbiota profiling, there are multiple sources of contamination including reagents and environment, which should be controlled for.
4. The dataset includes four different sampling locations but the text and figures (specifically Figs 1 and 2) do not group them accordingly or make it visually easier for the reader to identify these for comparison.
5. Figures 8 and 9: consider moving this to the supplement and instead visualize abundance and consistency of OTU’s predicted to encode for key enzymes (in case this is not already represented in the heatmap, which is unclear, as I note directly there).
6. PCA leads authors to the conclusion that there is a core microbiota and that the communities were very similar. However, an analysis at the genus or OUT level, which can be ecologically significant, might show a different picture and or/expand this view.

Author Response
We would like to thank all comments of the reviewer to our manuscript. We took them in consideration while reviewing the main text. Please find in attach the reply to all your comments, and bellow the replies to your general considerations.
Kind regards
Summary
This manuscript provides bacterial community profiles from the gut of a woodlouse host species that has been less explored in comparison to other relatives. The data is obtained through amplicon sequencing targeting the bacterial 16S rRNA in samples collected in four different locations in Portugal. Based on this data, potential functional roles of the community members are predicted using PICRUSt2, highlighting the putative presence of digestive enzymes that are ecologically relevant for woodlice. Also, the manuscript reports on the presence and co-infection of three previously described bacteria (Wolbachia, Ricketsiella and Hepatoplasma) in the majority of the samples isopods. The data is useful as a first exploration of the microbiota associated to this species and its potential contributions to the host. I believe that some interpretations must be revised and/or clarified, and special consideration must be given to the predictive nature of the approach used for the functional aspect. Please find a few general points detailed below, as well as more specific comments noted directly on the pdf file, since no line numbering was provided.
Authors Reply: Thank you for your comments and we replied and addressed all concerns that were presented more specifically in the PDF file, during the reviewing of the manuscript.
General comments:
- As noted in several specific comments, I suggest the authors to tone down their conclusions on function and biotechnological potential. These are predictions, meaning that there is no direct evidence of the presence of the corresponding genes, their expression or their activity in the host. Also, this approach is based on taxonomic similarity, which in many cases does not correspond to functional or ecological similarity. While the data is useful, it is important that the wording reflects the limitations of this approach.
Replied on the PDF file.
- Please clarify what is meant by “coverage” in Supplementary Table 1 and, in connection to this, make sure that the assessment of full representation of the diversity in the samples is fair. The manuscript states that the rarefaction curves show saturation, but this is not the case (see specific comments on pdf).
Replied on the PDF file.
- Including negative controls would have been an important addition to this dataset. As now generally recognized in microbiota profiling, there are multiple sources of contamination including reagents and environment, which should be controlled for.
Reply: In fact, we did not used a negative control during this project, but in others were I had the experience of doing so, I never observed any true of note influence of the contaminants sequences of the downstream analyses, since they were not present in high abundances, and we removed during the analyses the OTUs with less than 10 sequences.
- The dataset includes four different sampling locations but the text and figures (specifically Figs 1 and 2) do not group them accordingly or make it visually easier for the reader to identify these for comparison.
Reply: Thank you for you note, and we enhanced and made this observation more friendly to the reader.
- Figures 8 and 9: consider moving this to the supplement and instead visualize abundance and consistency of OTU’s predicted to encode for key enzymes (in case this is not already represented in the heatmap, which is unclear, as I note directly there).
Reply: We moved Figures 8 and 9 to supplementary data.
PCA leads authors to the conclusion that there is a core microbiota and that the communities were very similar. However, an analysis at the genus or OUT level, which can be ecologically significant, might show a different picture and or/expand this view.
Reply: To dimmish the possibility of misleading information, we calculated the core microbiome by using microbiomeanalyst, and used the PCA results to compare the bacterial population profiles determined for all samples. We changed the main text and discussion accordingly.
